# First Characterization of Human Dermal Fibroblasts Showing a Decreased Xylosyltransferase-I Expression Induced by the CRISPR/Cas9 System

**DOI:** 10.3390/ijms23095045

**Published:** 2022-05-02

**Authors:** Bastian Fischer, Vanessa Schmidt, Thanh-Diep Ly, Anika Kleine, Cornelius Knabbe, Isabel Faust-Hinse

**Affiliations:** Institut für Laboratoriums- und Transfusionsmedizin, Herz- und Diabeteszentrum Nordrhein-Westfalen, Universitätsklinik der Ruhr-Universität Bochum, Georgstrasse 11, 32545 Bad Oeynhausen, Germany; vschmidt@hdz-nrw.de (V.S.); tly@hdz-nrw.de (T.-D.L.); ankleine@hdz-nrw.de (A.K.); cknabbe@hdz-nrw.de (C.K.); ifaust-hinse@hdz-nrw.de (I.F.-H.)

**Keywords:** xylosyltransferase-I, CRISPR/Cas9, myofibroblasts, skeletonization, extracellular matrix, proteoglycans

## Abstract

Background: Xylosyltransferases-I and II (XT-I and XT-II) catalyze the initial and rate limiting step of the proteoglycan (PG) biosynthesis and therefore have an import impact on the homeostasis of the extracellular matrix (ECM). The reason for the occurrence of two XT-isoforms in all higher organisms remains unknown and targeted genome-editing strategies could shed light on this issue. Methods: XT-I deficient neonatal normal human dermal fibroblasts were generated by using the Clustered Regularly Interspaced Short Palindromic Repeats (CRISPR)/CRISPR-associated proteins (Cas) 9 system. We analyzed if a reduced XT-I activity leads to abnormalities regarding ECM-composition, myofibroblast differentiation, cellular senescence and skeletal and cartilage tissue homeostasis. Results: We successfully introduced compound heterozygous deletions within exon 9 of the *XYLT1* gene. Beside *XYLT1*, we detected altered gene-expression levels of further, inter alia ECM-related, genes. Our data further reveal a dramatically reduced XT-I protein activity. Abnormal myofibroblast-differentiation was demonstrated by elevated alpha-smooth muscle actin expression on both, mRNA- and protein level. In addition, wound-healing capability was slightly delayed. Furthermore, we observed an increased cellular-senescence of knockout cells and an altered expression of target genes knowing to be involved in skeletonization. Conclusion: Our data show the tremendous relevance of the XT-I isoform concerning myofibroblast-differentiation and ECM-homeostasis as well as the pathophysiology of skeletal disorders.

## 1. Introduction

Xylosyltransferase (XT) isoforms XT-I and -II catalyze the rate-limiting step of the proteoglycan (PG) biosynthesis by the transfer of activated xylose, originated from the natural substrate Uridine diphosphate (UDP)-xylose, to a defined serine residue of an acceptor peptide. The xylosylation initiates the formation of a tetrasaccharide linker, subsequently connecting the PG core peptide with glycosaminoglycans (GAG). Due to their involvement in PG biosynthesis, the XT-isoenzymes play an important role concerning extracellular matrix (ECM) homeostasis.

Because elevated XT-I activities are associated with the manifestation of fibrotic diseases, XT-I activity is proposed to be a suitable biomarker of different fibrotic diseases and myofibroblast differentiation [1]. Myofibroblasts are cells, which differentiate from fibroblasts during pathological organ remodeling and wound healing [2]. It is known that this differentiation is stimulated by the cytokine transforming growth factor beta 1 (TGF-β1) [3]. Myofibroblasts are characterized by *de-novo* expression of alpha smooth muscle actin (α-SMA), which leads to an increased cellular contractility [4]. Furthermore, myofibroblasts produce excessive amounts of ECM-associated compounds, such as, for instance, collagens and PG [5]. 

XT-I and XT-II proteins are encoded by the *XYLT1* (16p12.3) and *XYLT2* (17q21.33) genes, respectively. Mutations within these genes potentially lead to the development of different PG-associated diseases, inter alia affecting cartilage and bone tissue homeostasis. Homozygous and compound heterozygous *XYLT1* mutations are known to be causative for skeletal dysplasia, whereby phenotypic characteristics vary concerning severity and include distinct facial abnormalities, joint laxity and short stature [6,7]. Mutations within the *XYLT2* gene are causative for the manifestation of the spondylo-ocular syndrome, leading to hearing and visual defects, bone fragility and osteoporosis [8,9].

While the model organisms *C. elegans* and *D. melanogaster* each only harbor one XT-isoform, all higher organisms express both XT-isoforms [10]. The reason for this phenomenon is hitherto unexplained and remarkable, as both isoforms are able to catalyze PG core xylosylation. To get a little closer to the answer, we recently developed a XT-I selective mass-spectrometric assay to sensitively determine isoform-selective enzyme activity for the first time [11]. We further conducted both a siRNA-mediated knockdown and a Clustered Regularly Interspaced Short Palindromic Repeats (CRISPR)/CRISPR-associated proteins (Cas) 9-mediated knockout of the *XYLT2* gene in already XT-I deficient human embryonic kidney (HEK293) cells to investigate whether these cells can compensate for a decreased expression of both XT-isoforms [12]. By revealing that only HEK293 cells expressing an at least partially functional XT-II enzyme are proliferative, our data emphasized the importance of both XT-isoforms concerning the cellular metabolism. Of note, we additionally showed that decreased *XYLT2* levels did not result in a compensatory induction of *XYLT1* mRNA levels. 

Building on these results, the aim of the present study was to evaluate the impact of a decreased XT-I activity inter alia on the ECM homeostasis and myofibroblast differentiation of neonatal normal human dermal fibroblasts (NHDF). To generate those cells, we conducted a CRISPR/Cas9-mediated knockout, targeting exon 9 of the *XYLT1* gene.

## 2. Results

### 2.1. Generation of CRISPR/Cas9 Based Neonatal NHDF-Cells Showing a Strongly Reduced XYLT1 Expression 

NHDF-cells were transfected with the PX458 all-in-one plasmid-vector [13] using Lipofectamine LTX. The vector contained a sgRNA-sequence specifically targeting a region within exon 9 of the *XYLT1* gene. According to the green fluorescent protein (GFP)-expression, observed by fluorescence microscopy, transfection-efficiency was about 30%, 24 h post-transfection (Figure 1a). FACS-sorting was conducted 48 h after transfection, to isolate GFP-positive and therefore PX458 expressing cells. For this, we first defined the fibroblast-population in a dotplot-diagram (Appendix A) and further sorted only cells showing high fluorescence-intensities (Appendix A). Two weeks after FACS-sorting, cells were diluted to generate single-cell clones. Cells were cultivated upon confluence and DNA was isolated to conduct a T7-endonuclease assay and sequencing analysis. By quantifying the intensity of the additional product-bands observed after treating DNA isolated from CRISPR-cells with T7-endonuclease, a mutation-rate of approximately 80% was determined by using ImageJ. As opposed to this, only a single product-band was observed when treating DNA isolated from wildtype (WT)-cells (Figure 1b). These results were confirmed by Sanger-sequencing (Appendix A). To separate alleles and therefore better characterize mutations introduced by the CRISPR/Cas9 system, TA-cloning was conducted. The sequence-alignment confirms that the Cas9-endonuclease generated a double-strand break three bases upstream of the protospacer adjacent motif (PAM)-sequence (blue frame) and within the sequence complementary to the designed gRNA (red frame) binding within exon 9 of the *XYLT1* gene (Figure 1c). As a result of error-prone DNA-repair mechanisms, two distinct compound heterozygous mutations were introduced, characterized by the deletion of one (allele 1) and five (allele 2) bases, respectively. Both mutations led to a frameshift (p. (Tyr617fs)), resulting in a premature stop-codon and consequently a truncated XT-I protein (Figure 1c,d). In the following, CRISPR/Cas9 treated cells are abbreviated as *XYLT1^−/−^* cells.

### 2.2. Characterization of Neonatal NHDF Cells Showing a Diminished XYLT1 Expression

#### 2.2.1. Transcriptional Level

To initially ascertain possible differences in basal mRNA expression levels, we conducted qPCR-analysis after incubating control- and *XYLT1^−/−^* fibroblasts in standard cell-culture medium containing 10% fetal calf serum (FCS). Compared to controls, *XYLT1* mRNA expression levels were 7.1-fold decreased in *XYLT1^−/−^* fibroblasts, whereas *XYLT2* expression was 1.4-fold increased (Figure 2a,b). Gene expression levels of myofibroblast marker Actin Alpha 2 (*ACTA2)* (5.8-fold), Syndecan 2 (*SDC2)* (3.0-fold) and Intercellular adhesion molecule 1 (*ICAM-1)* (7.8-fold) were all induced in knockout fibroblasts (Figure 2c,e,f). Interestingly, as opposed to control fibroblasts, basal mRNA expression levels of the PG Aggrecan (*ACAN)* were nearly not detectable in *XYLT1^−/−^* cells (Figure 2d). 

XT-I protein activities were determined by using a newly developed in-house XT-I selective mass-spectrometry (MS) test. Analysis revealed that intra- and extracellular XT-I activities were 2.9- and 6.1-fold diminished in *XYLT1^−/−^* fibroblasts under standard cell-culture conditions, respectively (Figure 2g,h).

We induced myofibroblast-differentiation by treating WT- and *XYLT1^−/−^* fibroblasts with 5 ng/mL of TGF-β1 one day after conducting a serum-withdrawal. For comparison, we cultivated cells in serum-reduced (0.1% FCS) medium in parallel. 

Compared to WT, *XYLT1* expression levels were 9.2-fold lower in XT-I deficient cells under serum-reduced conditions. TGF-β1-based inductive effects in *XYLT1*-expression were much stronger in WT (20.9-fold) than in *XYLT1^−/−^* fibroblasts (9.0-fold, Figure 3a). Interestingly, *XYLT2* expression levels of *XYLT1^−/−^* fibroblasts were 1.4-fold increased under serum reduced conditions when compared to controls and treatment with TGF-β1 resulted in a stronger inductive effect (1.1- and 1.4-fold, respectively, Figure 3b). Compared to WT, *ACTA2* gene expression levels were significantly (1.3-fold) higher in XT-I-deficient fibroblasts after serum-withdrawal. Nevertheless, *ACTA2* mRNA expression was more strongly induced in controls (27.9-fold), than in cells showing a reduced *XYLT1* expression (17.5-fold, Figure 3c). No mRNA-expression of the PG *ACAN* was detected when cultivating *XYLT1^−/−^* cells under serum-reduced conditions. As opposed to controls (3.5-fold induction), transcription-levels of *XYLT1* deficient fibroblasts were also not inducible by TGF-β1 supplementation (Figure 3d). Relative to controls, gene expression levels of the cell-surface PG *SDC2* were 6.6-fold higher in *XYLT1^−/−^* fibroblasts after serum-withdrawal. While *SDC2* mRNA expression levels were inducible in controls (3.4-fold), TGF-β1 supplementation led to 1.7-fold reduced *SDC2* mRNA levels in *XYLT1^−/^*^−^ fibroblasts (Figure 3e). Gene expression of the surface protein *ICAM-1* was 7.6-fold higher in knockout cells under serum-reduced conditions. While no inductive effect was observed for WT fibroblasts, TGF-β1 treatment led to 3.1-fold increased mRNA expression levels in fibroblasts showing reduced *XYLT1* levels (Figure 3f).

In controls, TGF-β1-treatment resulted in a 4.7- (intracellular, Figure 3g) and 2.1-fold (extracellular, Figure 3h) elevated XT-I activity, when compared to untreated cells. In contrast, cytokine-treatment led to a slightly reduced intracellular XT-I activity in *XYLT1^−/−^* cells (Figure 3g). Extracellularly, no effect was detected in these cells after TGF-β1 treatment (Figure 3h).

#### 2.2.2. Translational Level

Following up on gene expression results, we performed immunostaining of WT- and *XYLT1^−/−^* fibroblasts after using different cultivating conditions, to quantify expression of myofibroblast-marker α-SMA on protein-level. Compared to control fibroblasts, determined corrected total cell fluorescence (CTCF) was always higher within all considered conditions in XT-I deficient cells (Figure 4a,b). In detail, α-SMA expression was 5.0-, 2.7- and 1.5-fold increased under basal, serum-reduced and TGF-β1 induced conditions, respectively. α-SMA fluorescence was stronger reduceable in *XYLT1^−/−^* cells (9.0-fold) than in WT cells (4.9-fold) after conducting a serum-withdrawal. While TGF-β1 supplementation led to an increased α-SMA expression in both cells, inductive effects were stronger in WT (10.8-fold) than in *XYLT1^−/−^* (6.0-fold) cells. TGF-β1-treated cells developed a spindled- and stellate morphology, which is characteristic for myofibroblasts (Figure 4a). 

In addition to immunostaining, we performed Western blot-analysis to quantify α-SMA protein expression under basal cell-culture conditions. In accordance with the previously determined transcriptional data, protein expression levels of α-SMA were 4.1-fold stronger in *XYLT1^−/−^* fibroblasts, characterized by intensive, sharp bands corresponding to a molecular weight of approximately 42 kDa (Figure 4c).

We also determined collagen type I alpha 1 chain (COL1A1) protein expression, which was decreased 1.9-fold in *XYLT1^−/−^* cells under basal conditions (Figure 5a,b). When compared to controls, extracellular TGF-β1 concentration, determined within the supernatant of cells, was 1.5-fold induced in XT-I deficient cells (Figure 5c).

### 2.3. Wound-Healing

To further compare myofibroblast-differentiation of controls and *XYLT1^−/−^* fibroblasts, we performed a wound-healing assay. Thereby, cells were cultivated under both serum-reduced- and TGF-β1-induced conditions for a total of 72 h (Figure 6).

The wound-healing assay revealed that controls as well as *XYLT1^−/−^* fibroblasts reached confluence between 36 and 48 h after introducing an artificial gap within the cell-monolayer. Thereby, WT cells filled the wound slightly faster than XT-I deficient cells under both cultivation conditions. It is noticeable that TGF-β1-treatment led to an initially enhanced migration of both controls and *XYLT1^−/−^* cells. 

### 2.4. Cellular Senescence 

To analyze and compare cellular-senescence of WT- and *XYLT1^−/−^* cells, we conducted a quantitative senescence assay based on the catalytic activity of the senescence-associated (SA) enzyme β- galactosidase (Figure 7). In controls, fluorescence was 1.1-fold elevated during serum-withdrawal when compared to cultivation under basal conditions. After treating cells with TGF-β1, fluorescence-signal and therefore cellular senescence became increased 1.2-fold in control cells. 

Relative to controls, cellular-senescence was continuously elevated during all tested cell-culture conditions in *XYLT1^−/−^* fibroblasts. Under basal culture conditions, cellular senescence was 1.9-fold induced in XT-I-deficient cells, when compared to controls. When cultivating CRISPR-treated cells in serum-reduced medium, fluorescence became 1.4-fold increased compared to the measured signal of cells cultivated under standard conditions. When conducting a serum-withdrawal, *XYLT1^−/−^* cells developed a 2.2- higher senescence than controls cultivated under the same conditions. Compared to cultivation under serum-withdrawal, TGF-β1-induction led to a reduced fluorescence-signal in *XYLT1* deficient fibroblasts (1.1-fold), which was nevertheless 1.2-fold higher than under basal cultivation conditions. 

### 2.5. Proliferation Capability 

To compare proliferation capability of WT- and *XYLT1^−/−^* fibroblasts, we conducted a WST-1 assay (Figure 8). The cells both proliferated most strongly under basal cell culture conditions. Serum starvation resulted in strongly reduced proliferation capacities of the cells in each case. TGF-β1 treatment led to different effects for both cells. In the XT-I-deficient cells, cytokine treatment resulted in an increased proliferation rate comparable to that under basal conditions. In contrast, WT fibroblasts showed strongly reduced proliferation values, which were even lower than those during cultivation under serum withdrawal.

## 3. Discussion

PG biosynthesis is composed of different enzymatic reactions, whereby the initial step is catalyzed by XT, of which two human isoforms (XT-I, XT-II) exist. So far, the reason for the existence of two XT isoenzymes has not been clarified and most publications do not differentiate between them. To further characterize the relevance of the XT-I isoform, we here performed an isolated CRISPR/Cas9-based *XYLT1* knockout in dermal fibroblasts. Thereby, we successfully introduced compound heterozygous mutations within exon 9 of the *XYLT1* gene resulting in frameshift mutations (p. (Tyr617fs)) causing a premature stop-codon and leading to a truncated XT-I protein. For the first time, we subsequently isolated and characterized *XYLT1^−/−^* dermal fibroblasts to better access the functional relevance of the XT-I isoform. 

Our data reveal altered expression levels of different ECM-associated targets in XT-I deficient dermal fibroblasts. As expected, *XYLT1* expression levels (Figure 2 and Figure 3a) as well as intra- and extracellular XT-I activities (Figure 2 and Figure 3g,h) were strongly diminished in *XYLT1^−/−^* fibroblasts under all considered cell-culture conditions. Previous studies showed that treatment of NHDF with the cytokine TGF-β1 leads to a strong induction of both *XYLT1* mRNA levels and XT-activity [14,15]. Compared to controls, *XYLT1* gene-expression was inducible to a much lower extent when treating cells with TGF-β1. The reason an inductive effect is observed at all in XT-I deficient cells at the transcript level is because the primer-pairs used bind upstream of the mutation-site within exon 6, resulting in a partial amplicon. In contrast, the XT-I activity, determined by using our recently developed isoform-selective MS-assay [11], is not inducible in these cells, indicating that the genetic mutations led to an extensive loss of function. 

Interestingly, mRNA expression levels of the XT-II isoform were elevated in cells showing a diminished XT-I expression (Figure 2b) and stronger inducible when compared to WT (Figure 3b). Earlier studies concerning other proteins have already showed that a diminished expression of one isoenzyme is often compensated by an elevated expression of the remaining isoform. For example, the deficiency of one or two murine cyclin-D isoforms leads to a compensatory induction of the third isoform. Solely the knockdown of all three cyclin-D isoforms results in a pathologic phenotype [16]. Further studies are needed to elucidate the relevance of two human XT-isoenzymes. In this context, our group has most recently been able to show that XT-deficient HEK cells are not viable [12].

As α-SMA is a widely accepted myofibroblast-marker [2], altered amounts of this protein indicate abnormalities in the differentiation of fibroblasts to myofibroblasts. Our gene expression data show a strongly increased *ACTA2* expression in *XYLT1^−/−^* fibroblasts (Figure 2c), which is inducible to a similar extent when compared to controls (Figure 3c). By conducting immunostaining and Western blot analysis, these observations were verified at the translational level (Figure 4). This could be explained by the observed increased expression of the cytokine TGF-β1 in the XT-I deficient fibroblast (Figure 5), which has previously been shown to induce α-SMA expression in various cell types [17,18]. TGF-β1 protein expression, in turn, might be increased in compensation to counteract a decreased myofibroblast differentiation capability caused by the introduced mutation. Different studies reveal that the cytokine is a key mediator of this transition [19,20] as well as a regulator of the *XYLT1* expression [21]. Our data show such an altered myofibroblast differentiation in the wound-healing assay. Under physiological conditions, activated myofibroblasts migrate towards the site of the wound and initiate an inflammatory response. The XT-I deficient fibroblasts initially close the artificial wound much slower than the controls, even after TGF-β1 induction (Figure 6). 

A highly important finding of our study is that, as a result of the *XYLT1* knockout, mRNA expression levels of the PG *ACAN* is no longer detectable and also not inducible in *XYLT1^−/−^* fibroblasts (Figure 2 and Figure 3d). A previous study already revealed reduced PG levels because of a disrupted Xylt1 activity in a mouse mutant suffering on dwarfism [22]. Other mouse-models furthermore showed an explicit association between dwarfism and decreased aggrecan expression [23]. Dwarfism is a type of a dysplasia primarily affecting the bone- and skeletal tissue [24]. Diseases affecting the skeleton are in general very heterogeneous concerning the underlying genetic cause and the resulting pathogenic phenotype. Skeletal dysplasia, as one type of such diseases, is subdivided into 450 different subtypes, whereby the molecular basis is mostly known [1]. In this context, recent studies showed *XYLT1* mutations to be causative for the manifestation of skeletal dysplasias [6,7,25], which could at least partially explain the reduced *ACAN*-expressions observed. 

As opposed to *ACAN*, basal gene-expression levels of the PG *SDC2* were upregulated in *XYLT1^−/−^* fibroblasts (Figure 2e). An elevated *SDC2* synthesis was previously shown in the context of different diseases, such as rheumatoid arthritis (RA), cardiovascular disease (CVD) and Chronic obstructive pulmonary disease (COPD) [26]. The manifestation of RA is generally associated with altered cartilage function inter alia characterized by abnormal PG biosynthesis and therefore directly related to XT-function [27]. Interestingly, previous publications have already shown a relationship between altered XT expressions and arthritis [28,29]. In addition, some cytokines, such as distinct interleukins and TGFβ, seem to induce *SDC2* expression. Therefore, the increased *SDC2* levels might also be related to the basally increased TGFβ1 expression. Notably, our data show decreased *SDC2* expression levels after treating *XYLT1^−/−^* fibroblasts with TGFβ1 (Figure 3e). This effect was not observed in WT cells. It is known that the SDC2 core protein actively binds TGFβ, which triggers a cascade resulting in an upregulated TGFβR expression and subsequently enhances its signaling [30]. A possible explanation of our observation therefore could be a TGFβ1-excess after supplementation of the cytokine.

The mRNA-expression level of the intercellular adhesion protein *ICAM-1* was strongly overexpressed in *XYLT1^−/−^* fibroblasts under basal cell-culture conditions as well as after conducting a serum withdrawal (Figure 2f). Importantly, *ICAM-1* expression was significantly inducible by TGFβ1 treatment (Figure 3f) only in XT-I deficient cells, but not in controls. Such an inductive effect was already shown for endothelial cells by Suzuki et al. in 1985 [31]. ICAM-1 is known to harbor a key role regarding the regulation of cellular responses in inflammation [32] and previous studies additionally show an induction of ICAM-1 expression by (pro)inflammatory cytokines, such as TNFα, IL1β and IFNγ [33]. Of note, a study by Clark et al. observed a correlation between increased ICAM-1 expressions and a cytoskeletal reorganization [34].

We also found that *XYLT1^−/−^* fibroblasts show a higher cellular-senescence within all tested conditions of cultivation. Cellular senescence is induced to inhibit the replication of harmed cells. During the final phase of wound-healing, myofibroblasts exhibit an elevated cellular senescence to avoid a manifestation of fibrotic tissue remodeling. Skeletal dysplasias were previously already associated with cellular-senescence. Patients suffering from Hutchinson-Gilford Progeria Syndrome (HGPS), which is accepted as a unique skeletal dysplasia [35], exhibit short-statue, show a diminished ECM-production and a premature senescent phenotype [36]. An elevated cellular senescence could also directly result from the introduced, CRISPR/Cas9-based mutations. Genetic alterations can potentially cause permanent cell-cycle arrest, which in turn stimulates cellular senescence [37]. 

To detect potential alterations regarding the proliferation capacity of the cells, we subsequently performed a WST-1 assay. The highest absorbance values were detected under basal conditions for both WT and XT-I deficient cells. As previously already shown for different cell lines [38,39], serum starvation also resulted in a decreased proliferation rate of both WT and *XYLT1^−/−^* fibroblasts. Interestingly, TGF-β1 treatment led to opposite effects in XT-deficient cells and controls. Control fibroblasts showed decreased proliferation rates, which were even lower than under serum withdrawal. This was to be expected and is in accordance with the results of several previous publications [40,41]. To the best of our knowledge, our data show for the first time an increased proliferative capacity of cell cultures after TGF-β1 stimulation. The reason for this cannot be conclusively clarified. A possible approach for follow-up studies could be to investigate SMAD-pathways in *XYLT1^−/−^* fibroblasts, as intracellular SMAD-proteins are known to be critical for signal-transmission to the nucleus triggered by the cytokine TGF-β1. In this context, Rich et al. have already proven that TGF-β1 suppresses cell proliferation in a Smad3-dependent manner [42].

## 4. Materials and Methods

### 4.1. Cell Culture

NHDF were purchased from Coriell (Camden, NJ, USA). Basal measurements were conducted by cultivating cells in standard cell culture medium, which consists of Dulbecco’s modified Eagle´s medium (ThermoFisher, Waltham, MA, USA) supplemented with 10% fetal calf serum (FCS, PAN-Biotech, Aidenbach, Germany), 2% L-glutamin and 1% antibiotic/antimycotic solution (100×, PAA, Pasching, Austria). Cells were cultivated at 5% CO_2_ and 37 °C. 

### 4.2. CRISPR/Cas9 Workflow

#### 4.2.1. Transfection of NHDF-Cells with a CRISPR/Cas9 All-in-One Vector 

For CRISPR/Cas9-based *XYLT1* knockout, 1 × 10^5^ NHDF cells were seeded onto a 12-well cell culture dish. 24 h later, cells were transfected (Lipofectamine LTX, Thermo Fisher Scientific) with 1.5 µg of all-in-one plasmid pSpCas9(BB)-2A-GFP (PX458) [13], containing a cloned sgRNA-sequence targeting exon 9 of the human *XYLT1* gene. The vector was a gift from Feng Zhang (Addgene plasmid #48138; http://n2t.net/addgene:48138 (accessed on 3 April 2022); RRID: Addgene_48138). 24 h post-transfection, transfection-efficiency was controlled by fluorescence microscopy. 

#### 4.2.2. Isolation of GFP-Positive NHDF Cells Using FACS-Technology

Using a S3e Cell sorter device (BioRad, Hercules, CA, USA), FACS-sorting was conducted 48 h after transfection, to isolate GFP-positive and therefore PX458-expressing cells. For this, we first defined the fibroblast-population in a dotplot-diagram and further sorted only cells showing high fluorescence-intensities. Two weeks after FACS-sorting, cells were diluted to generate single-cell clones. Cells were cultivated upon confluence and DNA was isolated to conduct sequencing analysis and a T7-endonuclease assay.

#### 4.2.3. T7 Endonuclease Assay

T7 endonuclease assay (NEB, Ipswich, MA, USA) was conducted according to the manufacturers protocol. Briefly, isolated DNA of untransfected (WT) and CRISPR/Cas9-transfected cells was amplified within the genomic region complementary to the gRNA used (*XYLT1*, exon 9). Due to a heteroduplex-formation, several product-bands were identified when incubating amplicons of transfected cells with T7-endonuclease. By quantifying the intensity of the additional product-bands, a mutation-rate was determined using ImageJ.

#### 4.2.4. TA-Cloning

To separate alleles and therefore better characterize heterozygote mutations introduced by the CRISPR/Cas9 system, TA-cloning was conducted according to the manufacturers protocol (ThermoFisher, Waltham, MA, USA). After replication of the circular vector using E.coli Top10, prokaryotic cells were lysed, plasmid-DNA was purified and subsequently analyzed by sanger sequencing.

### 4.3. Treatment of Dermal Fibroblasts with TGF-β1

To induce myofibroblast-differentiation, cells were treated with TGF-β1. In brief, 50 cells/mm^2^ were seeded in a 100 × 22 mm cell culture dish (Greiner Bio-One, Kremsmünster, Austria) and cultivated in standard cell culture medium for 24 h. After this, cells were washed with 1× phosphate-buffered saline (PBS) (ThermoFisher, Waltham, MA, USA, pH 7.4) and a serum-withdrawal from 10% FCS to 0.1% FCS was performed to reduce ECM-synthesis. This is an essential step in order to be able to attribute subsequently observed effects primarily to *TGF-ß1* induction. After an incubation of another 24 h, medium was either replaced by fresh serum-reduced medium supplemented with an adequate volume of 1× PBS (vehicle), or 5 ng/mL TGF-β1 (Miltenyi-Biotech, Bergisch Gladbach, Germany). In both cases, total volume of culture-medium was set to 6 mL. Cells were harvested after 48 h or 72 h to determine mRNA-expression or XT-activity, respectively. 

### 4.4. Nucleic Acid Extraction and Reverse Transcription

Cells were lysed by using RA1-buffer (Macherey-Nagel, Düren, Germany). Total RNA was isolated by using the RNA Spin Blood Kit purchased from Macherey-Nagel. The method was performed according to the manufacturer’s instructions. After the first purification step, we collected a 50 µL aliquot for subsequent DNA normalization. DNA extraction was performed according to the manufacturer protocol (NucleoSpin Blood Kit, Macherey-Nagel). Nucleic acid concentrations and purities were determined by using NanoDrop 2000 (ThermoFisher, Waltham, MA, USA). For reverse transcription, we used SuperScript II Reverse Transcriptase (ThermoFisher, Waltham, MA, USA), to transcribe 1 µg RNA to cDNA. 

### 4.5. Quantitative Real-Time Polymerase Chain Reaction (qPCR)

qPCR was performed as previously described [12]. In brief, 1 µg RNA was transcribed to cDNA using SuperScript II Reverse Transcriptase (Thermo Fisher Scientific, Waltham, MA, USA). Afterwards, 2.5 µL of diluted (1:10) cDNA, 0.25 µL of each respectively primer-pair (Table 1), 2 µL water and 5 µL SYBR green master-mix (Roche, Mannheim, Germany) were mixed and incubated (95 °C, 5 min.) to subsequently analyze gene expression by qPCR. After that, 45 cycles of the following repetitive program were performed: denaturation (95 °C, 10 s), annealing (see Table 1 for optimal annealing temperature, 15 s), and elongation (72 °C, 20 s). Finally, a melting curve analysis was performed. Relative transcription levels were determined in triplicate and calculated by the delta-delta Ct-method [36]. Normalization factor calculation was based on the geometric mean of the expression levels of Hypoxanthine Guanine Phosphoribosyltransferase (*HPRT*), Glyceraldehyde-3-phosphate dehydrogenase (*GAPDH*) and β2-Microglobulin (*β2M*).

### 4.6. Mass Spectrometric XT-I Assay

To selectively determine XT-I activity in cell-supernatants (extracellular activity) and lysates (intracellular activity), we used an in-house mass spectrometric (MS) XT-I assay. Cells were first cultivated as described before. Cell-lysis was then performed by incubating the cells with 750 µL of fresh lysis buffer (50 mM TRIS, 150 mM NaCl, 1% (*v/v*) Nonidet P-40, pH 7.8) for 10 min at 4 °C. Cells, which remained attached, were mechanically detached by using a cell scraper. Lysates were transferred to a 1.5 mL micro-centrifuge tube and centrifuged for 10 min at 10.000× *g* and 4 °C. Supernatants were collected and stored at −20 °C until use. 

For quantification, we used a previously described MS XT-test [43], which we modified by partially regarding the used acceptor-peptide and the incubation time of the reaction mixture. In detail, we incubated the reaction mixture for a total of 24 h (instead of 12 h) and used an acceptor-peptide, which is more XT-I selective [44], than the previous one. The assay is based on the transfer of xylose from UDP-xylose to the specific acceptor-peptide. Separation of the product was performed by using a reverse phase chromatography column (UPLC BEH C18, 1.7 mm). The UPLC-system was coupled to a tandem-mass spectrometer (XEVO TQ-S, Waters, Milford, MA, USA). By using specific standards, which contained the xylosylated peptide in different concentrations, we quantified the amount of xylosylated peptide within our probes (peak area), which correlated to XT-I activity. The peak area was finally normalized to cellular DNA. 

### 4.7. Quantification of Cellular Senescence 

To quantify cellular-senescence of control- and XT-I-deficient fibroblasts, we conducted a fluorescence-based assay as previously described [45]. Quantification is based on the hydrolyzation of the fluorogenic substrate 4-Methylumbelliferyl β-D-galactopyranoside (MUG, Sigma, Kawasaki, Japan) by the senescence-associated (SA) enzyme β-galactosidase to 7-hydroxyl-4-methylcoumarin, which is detectable spectrophotometrically (excitation: 360 nm, emission: 465 nm, integration: 40 µs). Fluorescence signal was normalized to protein-concentrations in the supernatants quantified by a BCA-assay.

### 4.8. Quantification of α-SMA Protein Expression by Immunostaining and Western Blot 

For immunostaining, cells were seeded in a density of 50 cells/mm^2^ in a 12-well cell culture plate (Greiner Bio-One, Kremsmünster, Austria). As described before, cells were either cultivated under standard conditions (10% FCS), under serum-reduced conditions (0.1% FCS) or induced with TGF-β1. Cells were cultivated for a total of 120 h after induction. Further steps were carried out as previously described [1]. As the primary antibody, we used a mouse-anti-human α-SMA antibody (Dako, Hamburg, Germany), diluted 1:50 in 1% BSA dissolved in 1 × PBS. To quantify α-SMA-fluorescence of control- and *XYLT1^−/−^* fibroblast, we calculated the corrected total cell fluorescence (CTCF) by using ImageJ.

To determine basal α-SMA expression of control and XT-I deficient fibroblasts by Western blot, we seeded 50 cells/mm^2^ in 100 mm cell-culture dishes using standard cell culture medium. Cells were cultivated for a total of 72 h, lysed and protein concentrations were determined using a BCA assay. For each sample, 20 µg protein were separated by SDS-PAGE (8–16%, 2 h, 125 V) and subsequently transferred onto a polyvinylidene difluoride (PVDF) membrane. For Western blotting, we used a semi-dry electro-blotting apparatus and blocked nonspecific binding-sites with 5% milk powder (dissolved in phosphate-buffered saline, pH 7.4) for 1 h. The membrane was then rotary incubated with the same α-SMA antibody used for immunostaining (dilution: 1:2000) for 1 h. For detection, we used a peroxidase conjugated secondary antibody (horse-anti-mouse IgG-HRP polyclonal antibody). As a loading control, we used a primary antibody against GAPDH (dilution: 1:2000, mouse-anti-human, Cell-Signaling, Danvers, MA, USA).

### 4.9. Wound Healing Assay

Wound healing assay was performed to compare the ability of wound healing between control and XT-I deficient fibroblasts. For this, 1 × 10^6^ cells were seeded in a 60 × 15 mm cell culture dish (Sigma, Kawasaki, Japan) and cultivated for 24 h in standard cell culture medium. On the next day, an artificial gap was created within the cell-monolayer using a 100 µL pipette-tip. Cells were washed with 1× PBS, to remove detached cells. Cells were then cultivated in serum-reduced medium for another 24 h. To induce myofibroblast-differentiation, cells were treated with serum-reduced media supplemented with 5 ng/mL TGF-β1 or further cultivated in serum-reduced media. Cells were monitored for a total of 72 h using a JuLI BR live cell analyzer system (NanoEnTek, Seoul, Korea). Changes in confluence within the area, where we initially created the gap, were quantified by using ImageJ. 

### 4.10. WST-I Proliferation Assay

The WST-I assay is based on the conversion of tetrazolium (light red) to formazan (dark red, absorbance 450 nm) catalyzed by mitochondrial succinate-tetrazolium dehydrogenase. Catalysis occurs only in cells that have an intact respiratory chain. For the assay, 1.700 cells per well of a 96-well plate were seeded in 100 μL of standard cell culture medium and incubated over-night. Cells were then either induced with TGF-β1 or cultivated in standard medium (10% FCS) or under serum starvation (0.1% FCS). After 44 h, the assay was started by adding 10 μL WST-I/well (t0). From this point on, absorbance at 450 nm (reference wavelength: 690 nm) was determined hourly, over a period of 4 h, using the Tecan Reader infinite 200 Pro (Tecan, Crailsheim, Germany). For evaluation, a determined background value (medium only + WST-I reagent) was subtracted from the samples. 

### 4.11. Statistics

Statistical analyses were performed using GraphPad Prism 9.0 (San Diego, CA, USA). Data were analyzed using Mann–Whitney U test and are shown as +/− standard error of the mean (SEM), whereby *p*-values lower than 0.05 were declared as statistically significant. 

## 5. Conclusions

In conclusion, our data reveal a relationship between diminished XT-I expression levels, both on mRNA and protein level, and altered expressions of targets, known to be involved in the homeostasis of skeletal- and cartilage tissue. Additionally, our data show for the first time, that reduced *XYLT1* levels in dermal fibroblasts lead to an abnormal myofibroblast-differentiation. Previous studies analyzed myofibroblast characteristics after XT-induction. Abnormal differentiation of *XYLT1^−/−^* fibroblasts was characterized by strongly induced α-SMA expression on mRNA- and protein levels, as well as a diminished wound-healing capability. Furthermore, we determined altered mRNA- and protein expression levels of certain ECM-related targets and a simultaneous TGF-β1 induction as well as an elevated cellular-senescence in XT-I deficient fibroblasts. 

## Figures and Tables

**Figure 1 ijms-23-05045-f001:**
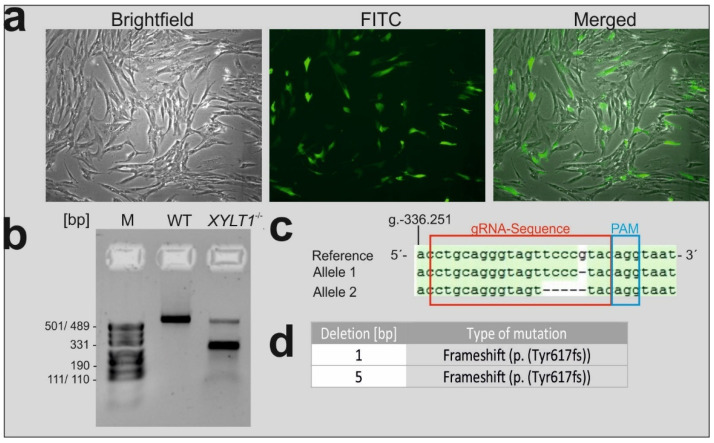
Transfection and subsequent characterization of CRISPR/Cas9-modified NHDF cells (*XYLT1^−/−^*). Based on the GFP-expression, efficiency of pSpCas9(BB)-2A-GFP(PX458) all-in-one vector transfection was about 30% (**a**). By conducting a T7-endonuclease assay, a mutation rate of 80% was determined based on the intensity of the additional (non-wildtype) DNA-band (**b**). Allele-specific sequencing analysis of the DNA isolated from transfected cells revealed compound-heterozygous mutations characterized by deletions of 1 (Allele 1) and 5 (Allele 2) base-pairs (bp) within the genomic region complementary to the gRNA-sequence targeting exon 9 of the human *XYLT1* gene (genebank accession ID: NG_015843.2). Resulting frameshift mutations (p. (Tyr617fs)) cause a premature stop-codon, leading to a truncated XT-I protein (**c**,**d**). M: marker; WT: wildtype.

**Figure 2 ijms-23-05045-f002:**
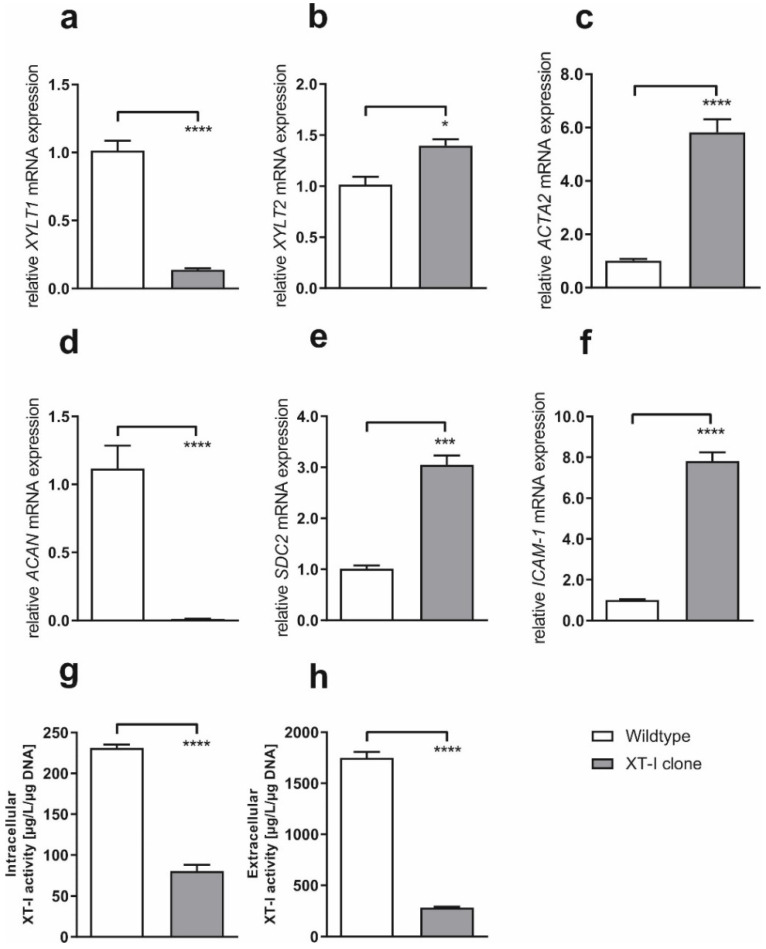
Relative basal mRNA expression levels of different ECM-related genes (**a**–**f**) and basal intra- and extracellular XT-I activities (**g**,**h**). Control- (n = 1) and *XYLT1^−/−^* fibroblasts (n = 1) were seeded (50 cells/mm^2^) and cultivated under standard cell-culture conditions. After 48 h of cultivation, total RNA was isolated and reverse transcribed. cDNA was used to determine relative mRNA gene expression levels of *XYLT1* (**a**), *XYLT2* (**b**), *ACTA2* (**c**), *ACAN* (**d**), *SDC2* (**e**) and *ICAM-1* (**f**) by real-time PCR. Data were normalized to a normalization factor, determined by calculating the geometric mean of *HPRT*, *GAPDH* and *β2M* mRNA expression levels, and expressed as a ratio to one cell line. Intra- and extracellular XT-I activities were determined after a cultivation-time of 72 h by mass-spectrometry and were normalized to DNA-concentrations. Values represent means ± SEM. * *p* < 0.05; *** *p* < 0.001; **** *p* < 0.0001.

**Figure 3 ijms-23-05045-f003:**
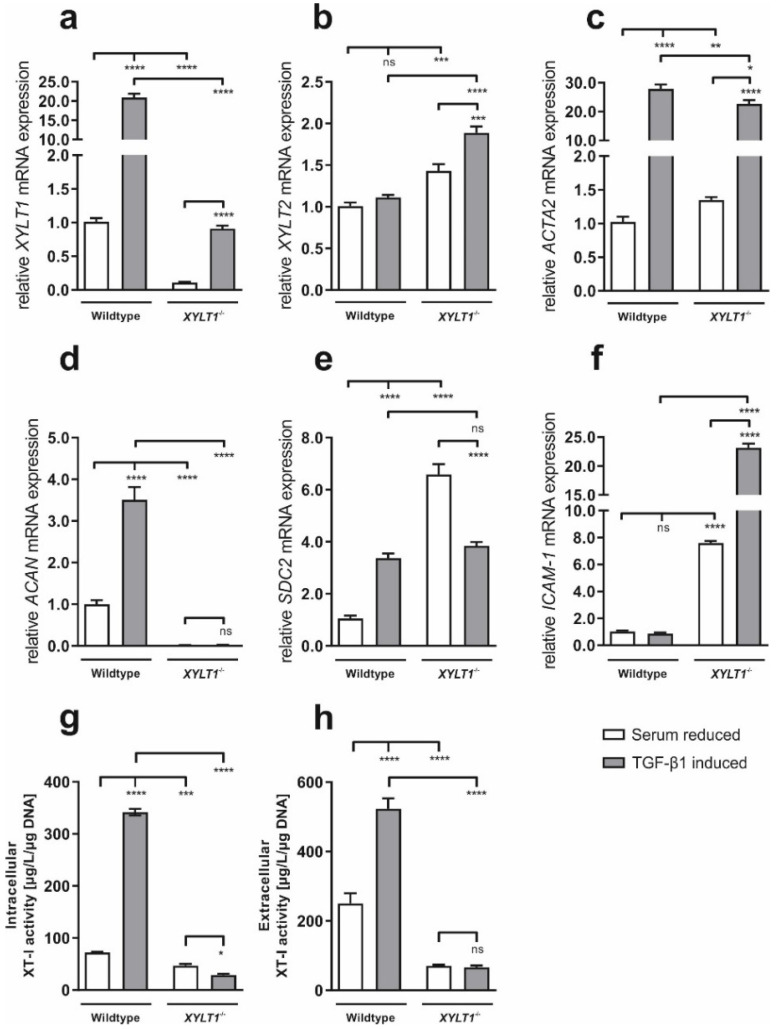
Relative mRNA expression levels of different ECM-related genes as well as intra- and extracellular XT-I activity after treating WT- and *XYLT1^−/−^* fibroblasts with TGF-β1. Wildtype and *XYLT1^−/−^* fibroblasts were seeded (50 cells/mm^2^) and incubated overnight. After 24 h, a serum withdrawal from 10% to 0.1% FCS was performed for 24 h. Thereafter, cells were incubated with vehicle (negative control) or TGF-β1 (5 ng/mL). After an additional 48 h, the total RNA was isolated and reverse transcribed. cDNA was used to determine relative mRNA gene expression levels of *XYLT1* (**a**), *XYLT2* (**b**), *ACTA2* (**c**), *ACAN* (**d**), *SDC2* (**e**) and *ICAM-1* (**f**) by real-time PCR. Data were normalized to a normalization factor, determined by calculating the geometric mean of *HPRT*, *GAPDH* and *β2M* mRNA expression levels, and expressed as a ratio to one cell line. Intra- (**g**) and extracellular (**h**) XT-I activities were determined after a cultivation time of 72 h by mass-spectrometry and normalized to DNA-concentrations. Values represent means ± SEM. ns: not significant; * *p* < 0.05; ** *p* < 0.01; *** *p* < 0.001; **** *p* < 0.0001.

**Figure 4 ijms-23-05045-f004:**
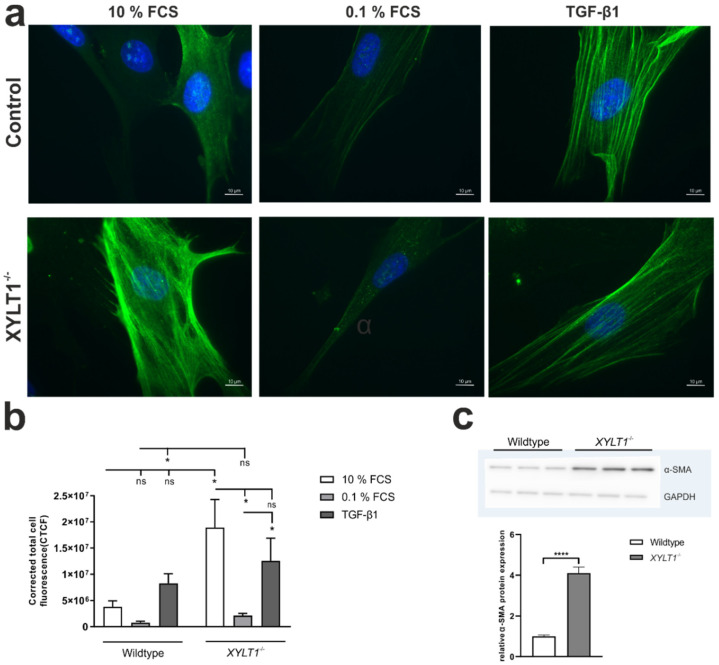
Quantification of α-SMA protein expression using immunofluorescence and Westernblot analysis. Wildtype and *XYLT1^−/−^* fibroblasts were seeded (50 cells/mm^2^) and incubated overnight. After 24 h, cells were further cultivated under standard cell-culture conditions (10% FCS) or a serum withdrawal from 10% to 0.1% FCS was performed for 24 h. Thereafter, cells were incubated with vehicle (negative control) or TGF-β1 (5 ng/mL) for another 120 h. Cells were then fixed and α-SMA protein expression was visualized by immunohistochemistry (**a**). Cell-fluorescence was quantified by using ImageJ to calculate CTCF (**b**). For Western blot analysis, cells were cultivated for a total of 72 h in standard-cell culture medium, containing 10% FCS. Cells were then lysed and protein-concentrations were determined using a BCA-assay. 20 µg protein/sample were separated by SDS-PAGE and subsequently transferred onto a polyvinylidene difluoride (PVDF)-membrane. Primary α-SMA antibody was diluted 1:2000. As a loading-control, we used a primary GAPDH-antibody (dilution: 1:2000). Intensities of protein-bands were quantified using ImageJ (**c**). Values represent means ± SEM. ns: not significant; * *p* < 0.05; **** *p* < 0.0001.

**Figure 5 ijms-23-05045-f005:**
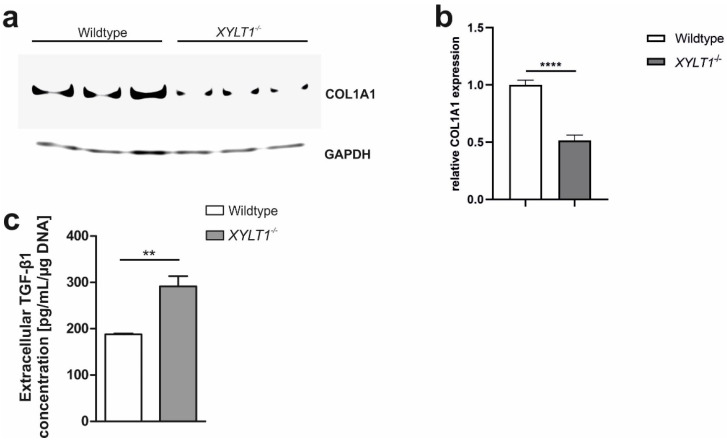
COL1A1- and TGF-β1 protein expression. For COL1A1 Western blot analysis, cells were cultivated for a total of 72 h in standard-cell culture medium, containing 10% FCS. Cells were then lysed and protein-concentrations were determined using a BCA-assay. 20 μg protein/sample were separated by SDS-PAGE an subsequently transferred onto a polyvinylidene difluoride (PVDF)-membrane. Primary COL1A1 antibody was diluted 1:2000 (**a**). As a loading-control, we used a primary GAPDH-antibody (dilution: 1:2000). Intensities of protein-bands were quantified using ImageJ (**b**). Extracellular TGF-β1 concentration was determined within the cell culture supernatant using a commercial ELISA test (**c**). ** *p* < 0.01; **** *p* < 0.0001.

**Figure 6 ijms-23-05045-f006:**
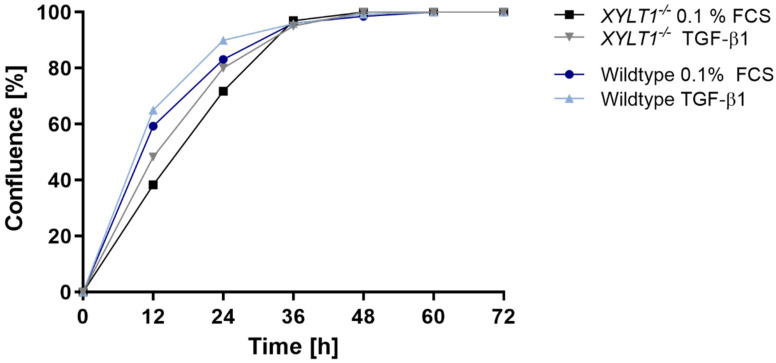
Wound-healing assay. To conduct a wound-healing assay, 1 × 10^6^ control and *XYLT1^−/−^* fibroblasts were respectively seeded in a cell culture dish and cultivated overnight. Thereafter, an artificial gap was created within the fibroblast-monolayer and a serum withdrawal from 10% to 0.1% FCS was performed for 24 h. Cells were subsequently incubated with a vehicle (negative control) or TGF-β1 (5 ng/mL) and wound-healing capability was analyzed for a total of 72 h using a live cell analyzer.

**Figure 7 ijms-23-05045-f007:**
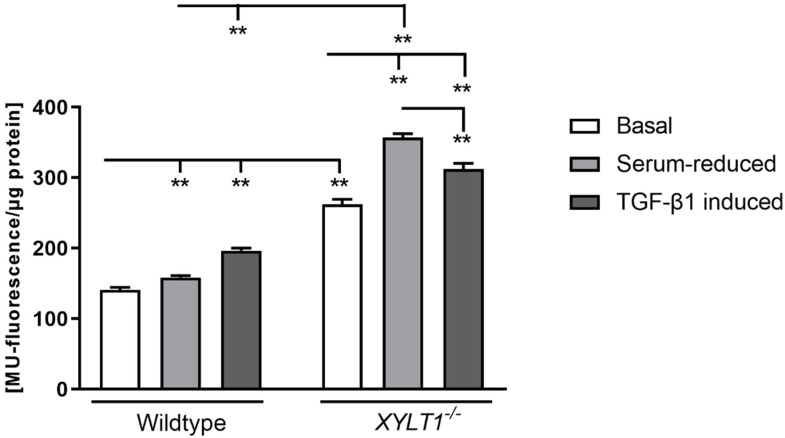
Quantification of the cellular-senescence of WT- and *XYLT1^−/−^* fibroblasts. Wildtype and *XYLT1^−/−^* fibroblasts were seeded (50 cells/mm^2^) and incubated overnight. After 24 h, cells were further cultivated under standard cell-culture conditions (10% FCS) or a serum withdrawal from 10% to 0.1% FCS was performed for 24 h. Thereafter, cells were incubated with a vehicle (negative control) or TGF-β1 (5 ng/mL) for another 72 h. Cells were lysed, centrifuged and supernatant was subsequently mixed with a reaction buffer, containing 1.7 mM 4-Methylumbelliferyl β- D- galactopyranoside (MUG). Spectrophotometrically measurements were conducted by using Tecan Reader Infinite 200 PRO (excitation: 360 nm, emission: 465 nm, integration: 40 µs). Values represent means ± SEM. ** *p* < 0.01.

**Figure 8 ijms-23-05045-f008:**
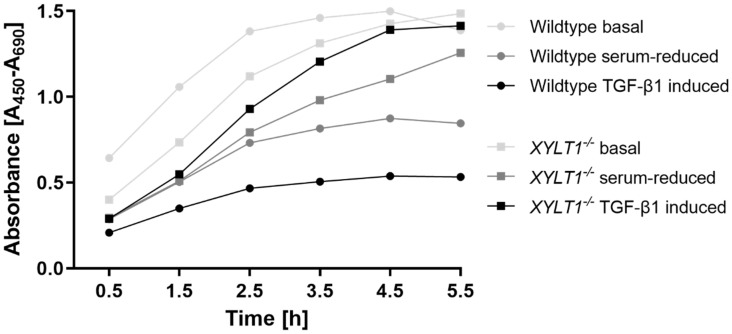
WST-1 assay to determine the proliferation capability of WT- and *XYLT1^−/−^* fibroblasts. Wildtype and *XYLT1^−/−^* fibroblasts were seeded, incubated overnight and further cultivated under standard cell-culture conditions (10% FCS) or a serum withdrawal from 10% to 0.1% FCS was performed for 24 h. Thereafter, cells were incubated with vehicle (negative control) or TGF-β1 (5 ng/mL) for another 72 h. After the addition of tetrazolium salt WST-1, absorbance was determined at six different time-points (0.5, 1.5, 2.5, 3.5, 4.5, 5.5 h post-supplementation). The absorbance measured correlates directly to the number of cells viable.

**Table 1 ijms-23-05045-t001:** Sequences and annealing-temperatures of the used quantitative real-time PCR primers.

Gene	Protein	5′-3′Sequence	Reference	T_A_ (°C)	Efficiancy
** *hACAN* **	ACAN	CACCCCATGCAATTTGAG GCCACTGTGCCCTTTTTA	NM_001135.4	63	1.92
** *hACTA2* **	α-SMA	GACCGAATGCAGAAGGAG CGGTGGACAATGGAAGG	NM_001320855.1	59	1.90
** *hβ2M* **	β2M	TGTGCTCGCGCTACTCTCTCTT CGGATGGATGAAACCCAGACA	NM_004048	63	1.98
** *hGAPDH* **	GAPDH	AGGTCGGAGTCAACGGAT TCCTGGAAGATGGTGATG	NM_002046	63	1.83
** *hHPRT1* **	HPRT1	GCTGACCTGCTGGATTAC TGCGACCTTGACCATCTT	NM_000194	63	1.94
** *hICAM-1* **	ICAM-1	ACCATCTACAGCTTTCCGGC CAATCCCTCTCGTCCAGTCG	NM_000201.3	59	1.91
** *hSDC2* **	SDC2	GGAGCTGATGAGGATGTA AATGACAGCTGCTAGGAC	NM_002998.3	59	1.90
** *hXYLT1* **	XT-I	TGTGACCTTCTCCACAGACG CCACGATGTGCTTGTACTGG	NM_022166.3	63	2.00
** *hXYLT2* **	XT-II	ACACAGATGACCCGCTTGTGG TTGGTGACCCGCAGGTTGTTG	NM_022167.3	63	1.95

## Data Availability

Not applicable.

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
