# Peer review of "First Characterization of Human Dermal Fibroblasts Showing a Decreased Xylosyltransferase-I Expression Induced by the CRISPR/Cas9 System"

_ijms, 2022, doi:10.3390/ijms23095045_

Round 1

Reviewer 1 Report

The presented manuscript is devoted to the role of one of the two isoforms of xylosyltransferases. The manuscript contains a large amount of experimental material, which goes beyond the evidence of the role of xylosyltransferase -1. This allowed the authors to reveal additional interesting facts. There is no doubt about the conclusions drawn from the results. However, there are some remarks in the design of the manuscript.
1. 41 - de nova-italic
2. 99 - figure 1, 128 - fig. 2 - bring monotonously
3.106 - genebank

4. figure 1 - no description of c and d
5. 141 - reduced serum - indicate how many times diluted
6. 140 - why did the experiment take place a day after serum sampling?
7. 232 - 10-6- is it quantity or concentration?
8. express level - no hyphen. Check if a hyphen is used frequently
9. Number the subsections in the sections Results and Materials and Methods
10. Make references according to the rules.

Author Response

We would like to thank you for your constructive comments and will address each of your comments separately (see attached word-file).

Reviewer 2 Report

This work describes the effects of decreased xylosyltransferase I expression on neonatal normal human dermal fibroblasts with the intent to further elucidate the reason for the occurrence of two xylosyltransferase isoforms in all higher organisms. Using the CRISPR/Cas9 gene editing technology the authors deleted functional copies of the XYLT1 gene and could show that such modified fibroblasts had a strongly reduced XYLT1 enzyme activity. In addition, they could show that gene-expression levels of several genes associated with the formation of the extracellular matrix were differentially regulated in the XYLT-1 deleted fibroblasts compared to the non-modified control cells. Notably they could show elevated alpha-smooth muscle actin expression in the XYLT1 deleted cells indicating an abnormal myofibroblast-differentiation in these modified cells. The presented data not only highlight the relevance of proteoglycan biosynthesis mediated by the Xyl-T1 isoenzyme for myofibroblast differentiation and the homeostasis of the extracellular matrix but also demonstrate the limited redundancy for the two xylosyltransferases of higher organisms.

This is a really nice paper presenting a very interesting set of data. My minor criticism concerns the use of abbreviations. All abbreviations should be explained in the text to enable the reader to understand the text without having to consult other sources of information.

The finding that XYLT1 deletion results in differential up-and downregulation of the two different proteoglycans ACAN and SDC2 is highly interesting and I would have liked to read a bit more of a discussion from a mechanistic/molecular perspective. How could it be explained on a molecular level that the lack of XylT 1 isoenzyme activity results in overexpression of one proteoglycan and downreguation of the other? Are these two proteoglycan precursors different targets for the two xylosyltransferase isoenzymes? Is there anything known about this in the literature?

Author Response

(The authors gave the same response as above.)

Reviewer 3 Report

I read the paper entitled: "First characterization of human dermal fibroblasts showing a  decreased Xylosyltransferase-I expression induced by the  CRISPR/Cas9 system" that shows and elegant demonstration of the rol of the enzyme on fibroblast´s differentiation and the expression of several relevant genes.

Some format and methods issues should be improved:

Abbreviation definitions:

line 68 NHDF  is not defined

Fig 1 legend: PAM is not defined

Format

To make the y axis scale equal for all graphs showing relative expression of mRNA, as well as, the ones for the enzymatic activities.

1 X PBS, should be indicated as  phosphate buffer saline, with a concentration and pH.

Methods

Explain the very different times frames  used for the incubation times for mRNA (48 h), immunostaining (120 h) and western blot (72 h).

Need to give more details about the qPCR, reference 12 (Fischer et al) does not give further details.  Need to explain why de delta Ct method was used.  Were the efficiencies of the PCRs of the studied and reference genes within a 5% of difference or was a Taqman system used?

Author Response

(The authors gave the same response as above.)
